# Collagen-Containing Fish Sidestream-Derived Protein Hydrolysates Support Skin Repair via Chemokine Induction

**DOI:** 10.3390/md19070396

**Published:** 2021-07-15

**Authors:** Ioanna Lapi, Ourania Kolliniati, Tone Aspevik, Eleftherios E. Deiktakis, Konstantinos Axarlis, Maria G. Daskalaki, Eirini Dermitzaki, Maria Tzardi, Sotirios C. Kampranis, Zouhir El Marsni, Katerina C. Kousoulaki, Christos Tsatsanis, Maria Venihaki

**Affiliations:** 1Laboratory of Clinical Chemistry, Medical School, University of Crete, 70013 Heraklion, Greece; iwanna_lapi@hotmail.com (I.L.); raliakolliniatis21@gmail.com (O.K.); el.deiktakis@gmail.com (E.E.D.); mol-grad392@edu.biology.uoc.gr (K.A.); m.daskalaki@med.uoc.gr (M.G.D.); renaderm@med.uoc.gr (E.D.); 2Institute of Molecular Biology and Biotechnology, Foundation for Research and Technology Hellas, 71100 Heraklion, Greece; 3Department of Nutrition and Feed Technology, Nofima AS, 5141 Bergen, Norway; tone.aspevik@nofima.no (T.A.); katerina.kousoulaki@nofima.no (K.C.K.); 4Laboratory of Pathology, Medical School, University of Crete, 70013 Heraklion, Greece; tzardi@med.uoc.gr; 5Section of Plant Biochemistry, Department of Plant and Environmental Sciences, University of Copenhagen, Thorvaldsensvej 40, 1871 Frederiksberg, Denmark; soka@plen.ku.dk; 6Seagarden AS, Karmsund Fiskerihavn, Husøyvegen 278, 4262 Avaldsnes, Norway; zouhir.el.marsni@seagarden.no

**Keywords:** collagen, fish sidestreams, chemokines, nutrition

## Abstract

Restoring homeostasis following tissue damage requires a dynamic and tightly orchestrated sequence of molecular and cellular events that ensure repair and healing. It is well established that nutrition directly affects skin homeostasis, while malnutrition causes impaired tissue healing. In this study, we utilized fish sidestream-derived protein hydrolysates including fish collagen as dietary supplements, and investigated their effect on the skin repair process using a murine model of cutaneous wound healing. We explored potential differences in wound closure and histological morphology between diet groups, and analyzed the expression and production of factors that participate in different stages of the repair process. Dietary supplementation with fish sidestream-derived collagen alone (Collagen), or in combination with a protein hydrolysate derived from salmon heads (HSH), resulted in accelerated healing. Chemical analysis of the tested extracts revealed that Collagen had the highest protein content and that HSH contained the great amount of zinc, known to support immune responses. Indeed, tissues from mice fed with collagen-containing supplements exhibited an increase in the expression levels of chemokines, important for the recruitment of immune cells into the damaged wound region. Moreover, expression of a potent angiogenic factor, vascular endothelial growth factor-A (VEGF-A), was elevated followed by enhanced collagen deposition. Our findings suggest that a 5%-supplemented diet with marine collagen-enriched supplements promotes tissue repair in the model of cutaneous wound healing, proposing a novel health-promoting use of fish sidestreams.

## 1. Introduction

Wound healing is a recovery response set in motion after tissue injury. This dynamic and complex process is a synchronized sequence of four interconnected stages; the hemostasis, the inflammatory, the proliferation, and the maturation phase. Upon skin injury, the development of a fibrin clot and coagulation occur, while leukocytes gradually infiltrate the wound to remove damaged tissue debris and foreign particles. Activated macrophages release several important growth factors and cytokines, setting up the ground for the next phase [1,2]. Distinct cell types present in the vicinity proliferate and migrate into the wound bed to produce extracellular matrix (ECM) components that form a contractile granulation tissue, replacing the original clot. In the final step of wound healing, balanced regulation between apoptosis and generation of new cells, as well as replacement of the type III collagen-rich granulation tissue with a type I collagen scar are critical for remodeling [3,4,5]. Chemokines are key orchestrators in each of these stages, predominantly contributing to promotion and inhibition of angiogenesis and recruitment of inflammatory cells [6,7].

For optimal wound healing, each phase has to take place in the right sequence and persist for a specific duration. If this tightly regulated cycle of events is disrupted, atypical skin restoration (fibrosis) or a non-healing chronic wound may arise [8,9,10,11]. Chronic wounds, which do not heal through the normal stages of healing in a predictable timeframe as acute wounds do, require long-term wound care [12,13]. These extended treatments impose a significant financial cost to the healthcare system, but also an immense humanistic burden [14,15]. It is a commonly known fact that wound healing is closely linked to nutrition, due to the high metabolic requirements of this repair process and the presence of nutrients that support cellular functions. Hence, nutrition substrates, both macro- and micro- nutrients, are essential in this reparative and regenerative procedure, as wounds need energy to fight infection and to produce collagen [16,17]. Even though nutritional supplementation was primarily used to protect against deficiencies of essential nutrients, current research has brought to light bioactive nutrients that prevent or treat diseases [18]. In particular, the beneficial impact of various compounds with high nutritional value that contain immune enhancing nutrients and metabolites have been extensively investigated in cutaneous wound healing models, suggesting that nutritional supplementation could be useful in wound treatment [19,20]. Nevertheless, the potential use of fish-derived extracts as nutritional supplements to support skin repair has not been widely evaluated to date.

The inflammatory phase of wound healing, in which the production of inflammatory mediators is elevated, is a fine-tuned procedure and a crucial step for skin repair. Over the past few years, most research has focused on the isolation of compounds with potential immunomodulatory properties from natural sources, especially from the biodiverse marine ecosystem [21]. A characteristic example is marine-derived collagen, which has shown pro-healing effects on the process of cutaneous repair [22,23]. A large amount of sidestreams, around 70–85% of the total weight of catch, is produced by the fish processing industry every year, significantly contributing to environmental pollution [22,24]. Although this raw material is generally considered waste, most of it contains great amount of muscle and connective tissue, high in protein content that may have important health-promoting properties. These proteins can be subjected to enzymatic hydrolysis, which is a mild processing technology that generates protein hydrolysates, a source of bioactive peptides [25,26,27]. In addition to their biological activity, many of these marine derivatives also have great nutritional value and, therefore, a beneficial effect on human health could emerge.

In the present study, we utilized fish sidestream-derived protein hydrolysates as dietary supplements to test their impact on the skin repair process using an in vivo cutaneous wound healing model.

## 2. Results

### 2.1. Chemical and Nutritional Analysis of Fish Sidestream-Derived Protein Hydrolysates

First, we sought to determine the chemical composition and nutritional quality of a series of sidestream-derived enzymatic protein hydrolysates produced by salmon and mackerel heads and backbones (Table 1). The analysis revealed that mackerel-based hydrolysates were rich in ash, compared with the salmon-derived hydrolysates, which were perceived as more salty tasting (Appendix A) [28]. This was also evident by monitoring the daily water consumption, as we have previously shown that mice fed with the hydrolysate based on mackerel heads, HMH, consumed more water compared to the other diet groups; yet no difference was observed on daily food consumption [29]. These protein hydrolysates also contain a wealth of amino acids and minerals, some of which are known to boost immune responses and promote health (Appendix A). In the context of wound healing, studies both in rodents and humans have suggested that arginine is essential for skin repair since its metabolites are important regulators of inflammation and collagen synthesis [30,31]. Similarly, zinc is a prominent immunomodulatory mineral that has been reported to play a major role in all phases of skin repair [32,33]. As shown in Appendix A, fish sidestream-derived collagen (Collagen), has the highest arginine-content, while the protein hydrolysate based on salmon heads (HSH) contains higher levels of arginine compared to the other hydrolysates and great amount of zinc.

### 2.2. Accelerated Wound Healing Is Observed in Collagen and HSH-Collagen Fed Mice

Next, we proceeded to evaluate the impact of the isolated fish extracts on the wound healing process; the protein hydrolysate produced from mackerel backbones (HMB) and HMH, salmon backbones (HSB) and HSH as well as Collagen and HSH-Collagen (Table 1). To simulate the average human supplement intake, mice were fed with 5% *w*/*w* supplementation of each extract, accounting that the concentration of a supplement should be around 5% of the daily human food intake. Soy protein was utilized in the control group given that the extracts contain a high amount of protein and the fact that the predominant protein source of the mouse food is soy. The wound closure for each diet group was monitored by measuring the wound area on days 0, 1, 3, and 5 (Figure 1A). With the exception of the groups receiving HSB and HMH as supplements, the wounds of all the other diet groups closed on day 1 in the same way as the control (Figure 1B). Interestingly, wound size in mice receiving supplement Collagen or HSH-Collagen was significantly reduced in the third post-wounding day compared to the corresponding control group. However, most diet groups reached the same wound size as the control on the last day of the experiment.

### 2.3. Collagen and HSH-Collagen Supplemented Diets Result in a More Mature Dermis after Wound Tissue Damage

In response to an acute wound injury, the protective skin barrier is disrupted and inflammatory cells are recruited into the damaged site, releasing important factors that are essential for the healing process. Accordingly, keratinocytes resident in the basal layer of epidermis become activated, migrate to the wound bed and initiate proliferating to fill the empty space within the injured area. We therefore performed histological analysis of the same wounds to confirm our macroscopic observations regarding wound closure. Wound tissues isolated on the 5th post-wounding day were stained with hematoxylin-eosin. As illustrated in Figure 2, *Collagen* alone or with HSH had a similar epithelisation pattern in contrast to the control group that presented hyperplasia in the stratified squamous epithelium. The control diet group showed a great number of inflammatory cells within the papillary layer of dermis, which is indicative of an intense inflammatory response that has not been resolved yet. All groups had a hyperkeratotic outer layer of epidermis, especially the control group. The excessive part of this layer would potentially be subjected to apoptosis and removed during the last stage of skin repair. It is also worth noting that the dermis of Collagen and HSH-Collagen supplemented groups appeared to be more fibrotic and collagenated compared to the corresponding control.

### 2.4. Collagen and HSH-Collagen Groups Show Increased Expression of Chemokines Involved in the Healing Process

We further investigated the potential mechanism of action of the supplements that were linked to accelerated wound closure, Collagen and HSH-Collagen. Despite the fact that chemokines are implicated in all four stages of wound healing, their role is most critical during the inflammatory and the proliferation phase. Hence, we examined whether the expression of CC- and CXC-chemokine subfamilies in the wound tissue differ between diet groups on day 5. Specifically, the mRNA expression levels of C-X-C motif chemokine ligand 1 (CXCL1), C-X-C motif chemokine ligand 2 (CXCL2), C-C motif chemokine ligand 3 (CCL3), and C-X3-C motif ligand 1 (CX3CL1) were determined (Figure 3). As presented in Figure 3A, the expression of CXCL1 and CXCL2 was highly enhanced in the *Collagen* group. HSH-Collagen supplementation also promoted an elevation in the levels of these chemokines, but to a lesser extent than Collagen alone. Moreover, both groups expressed high CCL3 transcripts in comparison to control mice. HSH-Collagen showed high expression of CX3CL1, while Collagen demonstrated a similar trend but without reaching statistical significance. On the contrary, diet groups such as HSB and HMH that did not have any positive effect on wound size; in fact these wounds appeared to close slightly slower that the control, showed too high chemokine expression, possibly indicating excessive inflammation at the wound site (Appendix A). Overall, a controlled and not excessive increase in the expression levels of CC- and CXC-chemokine subfamilies may facilitate healing. In addition to chemokines, pro-inflammatory cytokines, such as interleukin-6 (IL-6) and tumour necrosis factor-α (TNF-α), have critical roles in the inflammatory phase. Although the production of these cytokines is gradually reduced during the resolution of inflammation, IL-6 still participates in the repair and tissue regeneration process since its signalling accounts for the balanced switch to a reparative environment. We therefore measured IL-6 and TNF-α production at the wound site using the enzyme-linked immunosorbent assay (ELISA). A trend towards an increase in IL-6 levels was observed in Collagen and HSH-Collagen compared to the control group without being statistically significant (Figure 3B). In contrast, all groups clearly exhibited similar TNF-α levels in the wound tissue (Figure 3C).

### 2.5. Collagen-Supplemented Diet Stimulates VEFG-A Expression at the Wound Site

In addition to cytokines and chemokines, macrophages also release growth factors, such as transforming growth factor-β (TGF-β) and VEGF. These growth factors are important for angiogenesis, cellular proliferation and migration during the proliferation phase. As shown in Figure 4A, the expression levels of VEGF-A were significantly upregulated in the Collagen diet group, while slightly increased in HSH-Collagen. No difference was noted in TGF-β1 mRNA levels between the control and test groups. Similarly, secreted TGF-β1 levels were not affected in the Collagen or HSH-Collagen supplementation groups (Figure 4B). Although hypoxia-inducible factor-1 alpha (HIF-1a) positively regulates the wound healing process by stimulating the expression of these soluble growth factors, dietary supplementation with the extracts did not induce its expression (data not shown). 

### 2.6. Collagen Production Is Increased in HSH-Collagen Supplemented Group

During the proliferation phase, a variety of distinct cell types, such as endothelial cells, keratinocytes and fibroblasts, are recruited to the wound area, leading to the formation of the ECM-rich contractile granulation tissue. Since this temporary connective tissue is primarily composed of collagen, we sought to determine the collagen levels in the wound tissue on day 5 (Figure 5). Despite the fact that Collagen1a1 mRNA levels had no difference between diet groups, collagen deposition at the wound bed was enhanced. Specifically, collagen production was not increased in the *Collagen* group but was highly enriched in mice fed with *HSH-Collagen*, suggesting that there is increased collagen deposition rather than de novo expression or that increased collagen production is affected at a post-transcriptional level.

## 3. Discussion

In the present study, we used an in vivo cutaneous wound model to evaluate the effect of HSB, HMH, HMB, HSH, Collagen and HSH-Collagen on the wound healing process. HSH was selected to be combined with Collagen given prior evidence for its immunodulatory properties and the fact that it contains high amounts of zinc and arginine, both known to possess pro-healing properties [29,30,31,32]. According to the results, mice fed with Collagen and HSH-Collagen supplemented diets had accelerated healing that was apparent by a faster reduction in wound size on the third day post injury, in comparison with the other diet groups. Control mice were fed with normal diet plus 5% *w*/*w* soy protein given that the fish sidestream-derived extracts contain high amount of protein and the fact that soy represents 84% of total protein in normal chow diet. However, soy is estrogenic and some studies have reported that oestrogen reduces inflammation and affects wound healing, which could potentially mask the positive effects of the fish sidestream-derived supplements [34,35,36]. For the subsequent analyses, we focused on the two groups that showed more effective wound size reduction. Histological analysis revealed a more collagenated fibrous connective tissue present in Collagen alone or in combination with HSH-supplemented groups. On the other hand, soy supplementation that was used as the control diet still showed signs of inflammation within the papillary dermis in combination with hyperplasia in the stratified squamous epithelium, which may indicate a slower healing rate. We further explored potential changes in molecular events induced by these supplements, and for this reason, the expression and secretion of distinct mediators that have crucial roles in different aspects of the repair process were investigated. 

Chemokines are central contributors in the healing process, signalling immune cell recruitment and angiogenesis. Collagen supplementation led to elevated mRNA levels of CXCL2 and CXCL1 at the wound site. The mRNA expression of these chemokines was also high in the HSH-Collagen diet group, but less than in the Collagen group. CXCL1 and CXCL2 are released to activate and regulate the recruitment of inflammatory cells, neutrophils and macrophages, to the damaged wound region, and they are also potent angiogenic and granulation tissue-stimulating mediators [6,37]. A defective expression of these chemokines has been associated with impaired wound repair, most likely due to a reduction in neutrophil chemotaxis and angiogenesis [38,39,40,41]. On the other hand, sustained leukocyte infiltration, due to disturbed expression of these chemokines, can result in prolonged inflammation and subsequent delay in healing, as is the case of diabetic-related wounds [42]. Additionally, both groups presented increased expression of CCL3 compared with the corresponding control. Recruited neutrophils and macrophages release additional chemokines, such as CCL3, to enhance migration of inflammatory cells to the wound that release pro-angiogenic growth factors, which are important for neovascularization [7]. High expression levels of CCL3 has been linked with enhanced wound healing [43,44,45]. In contrast to Collagen, HSH-Collagen showed a statistically significant rise in CX3CL1 transcripts. Gene knockout studies have uncovered the importance of CX3CL1 in wound healing by unravelling its interaction with CX3CR1-expressing cells that stimulate the assembly and function of macrophages [46]. It is also worth noting that HSB and HMH supplementation was not effective on wound healing and resulted in higher levels of the aforementioned chemokines than those expressed in tissues from mice fed with collagen-containing supplements. This may be attributed to prolonged inflammation and subsequent inhibition of timely wound closure. An excessive chemokine response during the skin repair process has been linked to overactive inflammation, which could lead to inhibition of neovascularization and subsequent retention in the inflammatory phase, delaying wound closure or creating a hostile environment for the development of a nonhealing wound [47,48,49]. Overall, our results are consistent with previous reports showing that improvement in wound repair can be achieved by modulating CC- and CXC-subfamilies of chemokines. 

In addition to phagocytosing debris and bacteria, macrophages release growth factors, such as platelet-derived growth factor (PDGF) and VEGF, critical for promoting angiogenesis [5]. Angiogenesis is essential not only in the inflammatory phase to drive leukocyte migration into the wound region, but also during the proliferation phase to form new blood vessels that provide nutrients to meet the increased metabolic requirements of the proliferating cells [1]. Our findings indicate that the Collagen-supplemented diet induced an increase in VEGF-A levels, while HSH-Collagen administration showed a trend, but not statistically significant elevation, in VEFG-A expression. Collagen supplementation may promote healing via VEGF-A expression, which is known to enhance the formation of lymphatic capillaries in cutaneous wound repair [50]. Furthermore, TGF-β is also a pro-angiogenic factor and a potent inducer of endothelial, fibroblast, and keratinocyte cell proliferation and migration [51]. However, both diet groups did not affect TGF-β1 levels in comparison with the corresponding control. 

A wide variety of marine compounds are broadly utilized in pharmaceutical and biomedical applications due to their bioactivity, such as antimicrobial, antioxidant, and anti-inflammatory effects [21,52]. Over the past decades, peptides and proteins from different marine sources have been shown to possess immunomodulatory activities. We have previously reported that compounds isolated from marine algae present strong anti-inflammatory action [53,54]. It is therefore evident that multiple studies have explored the role of different marine-derived extracts in the context of wound healing. Various marine materials, such as wound dressings and small-molecule modulators have been shown to stimulate the repair process and reduce scar formation [55]. For instance, studies have demonstrated that collagen of different marine origin has promising effects on the process of wound repair. It is also a healthier choice to substitute mammalian with marine collagen since the former has been associated with pathological risks [22,23]. Even though an enormous number of by-products are generated from fisheries every year and discarded as waste, this raw material is rich in protein and, thus, can be used as a good source of bioactive compounds [56,57]. In the present study, we compared the chemical composition and nutritional quality of protein hydrolysates that were isolated from salmon and mackerel heads and backbones. Protein hydrolysates obtained from diverse marine sources contain small peptides that possess not only immunomodulatory activities, but also great nutritional value [21]. Our analysis illustrated that Collagen and salmon head-based hydrolysates contain high levels of protein. A great amount of protein is critical during healing since it provides the essential building blocks for the newly formed tissue. These extracts also have increased concentration of amino acids and minerals, some of which are known to enhance immune responses and promote skin repair. In particular, both Collagen and HSH have high levels of arginine, while HSH also contains a significant amount of zinc. 

Our results propose that the extracts, Collagen and HSH-Collagen, have a positive role in the reparative and regenerative process of wound healing when used as dietary supplements. Supplementation with these collagen-containing extracts, either Collagen alone or HSH-Collagen, led to an accelerated healing accompanied with upregulated expression of important chemokines. Collagen supplementation also modified the expression of the potent angiogenic factor VEGF-A. Although collagen mRNA synthesis was not induced, enhanced collagen deposition was observed in the wound tissue of the HSH-Collagen treated group, potentially due to post-transcriptional induction of collagen production or deposition of collagen fragments obtained through nutrition. According to earlier studies, type I collagen within the wound triggers the expression of a matrix metalloproteinase, collagenase-1, which in turn facilitates keratinocyte migration during reepithelialization [58,59]. The ingested collagen observed in our study after supplementation with the extracts could promote healing via a similar way of action. In agreement with our results, other studies have shown that oral administration of fish skin-derived extracts improved cutaneous wound healing in rodents by enhancing collagen deposition and angiogenesis. In particular, peptides enzymatically hydrolyzed from the skin of either chum salmon or marine fish have been reported to induce wound healing in rats following cesarean section [60,61]. Zhang et al. also found that oral administration of skin gelatin from chum salmon enhanced wound healing in diabetic rats [62]. Our findings show that both Collagen and HSH-Collagen supplementation similarly stimulated wound closure; however, differences in chemokine and VEGF-A expression as well as collagen deposition indicate that these two extracts possibly promote healing through distinct mechanisms. 

There are several limitations in the study. Analysis of wound closure was performed until day 5, when the wound was not fully healed to allow collection of tissue samples, therefore the timing of complete wound healing was not evaluated. In addition, samples were collected to evaluate expression of different factors at the tissue restoration stage and not at the early inflammatory stage of the healing process, restricting the information obtained on the effect of the different supplements. Moreover, the effect on chemokine expression was analyzed at the mRNA level, not allowing identification of a potential effect at the post-transcriptional level. Nevertheless, our findings clearly support the effect of collagen containing fish sidestream-derived supplements on skin homeostasis. 

In conclusion, nutritional supplementation with collagen-enriched fish sidestream-derived extracts appears to have beneficial effect on wound healing in the murine cutaneous wound model. 

## 4. Materials and Methods

### 4.1. Preparation of Fish Sidestream-Derived Nutritional Supplements

Enzymatic protein hydrolysates based on mackerel *(Scomber scombrus)* heads (HMH) and backbones (HMB) and salmon *(Salmo salar)* heads (HSH) and backbones (HSB) were produced according to Aspevik et al. [28]. Briefly, raw materials were milled and combined with tap water. For the enzymatic hydrolysis, mixtures were stirred vigorously before adding the enzyme (10 U/g protein). After 1-h hydrolysis at 55 °C, the temperature was raised to 90 °C to terminate the enzyme activity. The temperature was then lowered to 60 °C and the hydrolysate was placed into a 3-phase separating centrifuge. Enzyme inactivation at 90 °C was followed and the liquid phase was separated into the oil and aqueous phase. The aqueous phase was further filtered and concentrated on a 4-stage falling film evaporator before spray-drying. The resulted purified hydrolysates were hygroscopic, white to off-white, fluffy powders. Flounder skin collagen was kindly provided by Seagarden (Karmøy, Norway). Chemical and nutritional properties of these protein hydrolysates are shown in Appendix A and Aspevik et al. [28]. A concentration of 5% *w*/*w* in normal chow diet (4RF21, Mucedola, Settimo Milanese, MI, Italy) was used for each extract. Since these extracts contain a high amount of protein, and the fact that 84% of the total protein in normal diet is soy, we supplemented normal diet with 5% *w*/*w* of soy protein (94% protein content) and utilized it as the control diet group.

### 4.2. Wounding

Four to five male mice of C57BL/6 genetic background, 6–8 weeks of age, were kept under controlled conditions of light cycle (12 h day/night) and temperature (21–23 °C) prior to treatments. Animals were maintained in a pathogen-free animal facility in the Medical School of the University of Crete, Heraklion. All experimental protocols conducted in this study were in conformity with protocols approved by the Animal Care Committee of the University of Crete, School of Medicine (Heraklion, Crete, Greece) and the Veterinary Department of the Region of Crete under license number 269904 (Heraklion, Crete, Greece). 

Individually housed mice were randomly divided into seven groups and fed with normal diet plus supplements for two weeks before the initiation of the experiment. The day of the experiment, anesthesia was induced by intraperitoneal injection of a mixture of 120 mg/kg of body weight ketamine, 8 mg/kg xylazine and 1 μg/kg fentanyl and hair was removed. As depicted in Figure 1A, four excisional wounds were generated on their dorsal skin with a biopsy punch (3 mm in diameter). During the experiment, mice were fed with each dietary supplement. After wounding, photographs of wounds of all mice were taken on days 0, 1, 3, and 5. In addition, the margins of the wounds were traced via a permanent marker on a transparency paper. On the 5th day, animals were sacrificed and wound tissues were collected. Changes in wound size were calculated using ImageJ software and expressed as the percentage of the initial wound area. 

### 4.3. RNA Extraction and Real-Time PCR 

Wound tissues, which were collected on the last day of the experiment, were used for gene expression studies. For RNA extraction, wounds were homogenized in TRI Reagent (Sigma-Aldrich) and total RNA was isolated according to the recommended protocol. cDNA synthesis was performed with TaKaRa PrimeScript^TM^ RT reagent kit (Perfect Real Time) (RR037A, TaKaRa, Bio Inc, Kusatsu, Shiga, Japan), following manufacturer instructions. The expression profile of key mediators of the wound healing process was determined. In particular, mRNA expression levels of TGF-β1, VEGF-A, Collagen1a1, CXCL1, CXCL2, CCL3, and CX3CL1 were analyzed by quantitative PCR (real-time PCR). B-actin was utilized as the internal control. A list of the primers used is presented in Table 2. The reactions were performed in the qPCR machine (Applied Biosystems^®^, Foster City, CA, USA), while the mRNA transcription analysis was conducted via StepOne^TM^ software v2.3 and GraphPad Prism 8.0 software (GraphPad Software, San Diego, CA, USA).

### 4.4. Tissue Processing and Histological Analysis

After completion of the experiment, wound tissues were isolated, fixed in 10% formalin and gradually dehydrated in ethanol. The specimens were embedded in paraffin wax blocks and cut into 3 μm sections. After deparaffination, sections were stained with hematoxylin and eosin (H&E) staining. Microphotographs were taken in a light microscope and the stained sections were assessed for detecting morphological differences after dietary supplementation with distinct fish-derived extracts.

### 4.5. Soluble Collagen Assay

Newly synthesized collagen during wound healing was measured. According to the instructions of the Sircol^TM^ Soluble Collagen Assay (S1111, Biocolor Life Science Assays, UK), tissue extracts were incubated in 0.5 M acetic acid overnight at 4 °C in order for collagen to be released into solution. The Sircol dye reagent, which binds to the tri-peptide sequence (Gly-X-Y) of mammalian collagens type I to V, was added to the isolated collagen and the mixtures were placed at a mechanical shaker for 30 min. In parallel with the samples, bovine type I collagen was incubated to generate the collagen standard curve. The precipitated collagen was washed with the Acid-Salt Wash Reagent, which facilitated removing the unbound dye from the collagen-dye complex. After centrifugation at 12,000 r.p.m for 10 min, the collagen bound dye was released by the addition of the alkali reagent containing 0.5 M NaOH. The absorbance was measured at 555 nm using a microplate reader and the collagen concentration of the samples was calculated as microgram per milligram of tissue weight.

### 4.6. ELISA

The levels of TGF-β1 in the wounds were determined by the DuoSet ELISA kit (DY1679, R&D Systems, Minneapolis, MN, USA). Briefly, wounds were homogenized in Phosphate Buffered Saline (PBS) supplemented with protease inhibitors (A32965, Thermo Scientific, Waltham, MA, USA) and a 96-well plate was coated with the Capture Antibody, sealed, and incubated overnight. The next day, the plate was washed with Wash buffer (0.05% Tween 20 in PBS), and Block Buffer (5% Tween 20 in PBS) was added to block non-specific binding and reduce background. To activate latent TGF-β1 to the immunoreactive form, samples were treated with 1N HCl for acid activation, followed by incubation with 1.2 N NaOH/0.5 M HEPES for neutralization of the acidified samples. Accordingly, the standards and samples were added with the appropriate dilutions. Wells were then incubated with Detection Antibody, followed by the addition of Streptavidin-HRP solution. A washing step was required before the addition of each reagent. Finally, the plate was incubated with freshly mixed Substrate Solution and the reaction was stopped by adding Stop Solution (2N H_2_SO_4_). The microwell absorbance was read at 450 and 570 nm using a microplate reader and the results were analyzed via Microsoft Excel and GraphPad Prism 8.0 software. The protein concentration of each sample was measured by Bicinchoninic assay following manufacturer’s instructions (Sigma-Aldrich, St. Louis, MI, USA), and used for normalization. A similar procedure was followed for detecting IL-6 (431304, Biolegend, San Diego, CA, USA) and TNF-α (430905, Biolegend, San Diego, CA, USA) using ELISA. The measurement units used for quantification were picogram of each cytokine (TGF-β1/IL-6/TNF-α) per milligram of tissue protein for each sample.

### 4.7. Statistical Analysis

The results were expressed as mean ± standard deviation (SD). For multiple group comparisons, one-way analysis of variance (ANOVA) was applied for determining the significance (*p* < 0.05) differences. However, two-way ANOVA followed by Sidak’s post hoc test was employed to analyze the time-course curve for the wound healing process. Results were confirmed using Bonferroni and Holm-Sidak post hoc test. Statistical analyses were performed using GraphPad Prism version 8.0.

## 5. Conclusions

In conclusion, our findings show for the first time that dietary supplementation with fish sidestream-derived Collagen or the hydrolyzed protein containing HSH-Collagen results in accelerated wound healing in vivo as is evidenced by macroscopic evaluation of wound closure and histological analysis. These collagen-containing diets enhance skin repair via inducing the expression of CC- and CXC-family chemokines and the potent angiogenic factor VEGF-A. In addition, they are also associated with increased collagen levels in the repaired tissue, suggesting high collagen deposition. Therefore, the results indicate a novel use of collagen-containing dietary supplements of marine origin in promoting skin repair.

## Figures and Tables

**Figure 1 marinedrugs-19-00396-f001:**
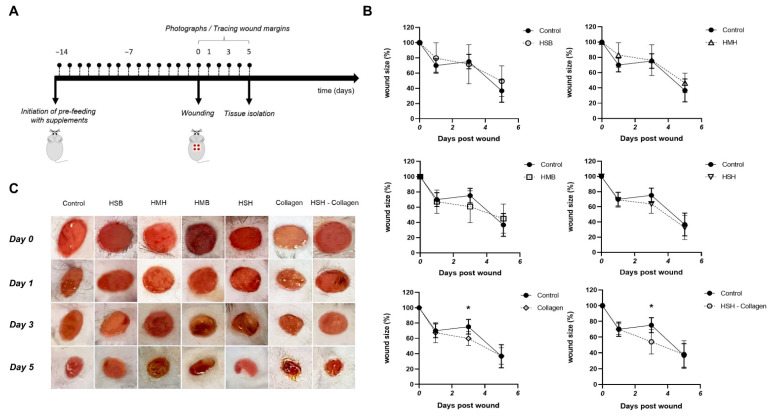
Collagen and HSH-Collagen supplementation accelerated wound closure. Changes in wound size are expressed as the percentage of the initial wound area. (**A**) Schematic diagram of the experimental protocol. After a 2-week period of pre-feeding with normal diet plus supplements, wounding was performed. The wound area was calculated by taking photographs and tracing the wound margins on the indicated post-wounding days. Wound tissues were collected on the 5th day post injury. (**B**) Macroscopic evaluation of wound closure on the indicated post-wounding days. (**C**) Representative photographs from all diet groups showing wound size reduction on different days post injury. Two-way ANOVA followed by Sidak’s post hoc test was employed to analyze the time-course curve. Values are expressed as the mean ± SD. * *p* < 0.05.

**Figure 2 marinedrugs-19-00396-f002:**
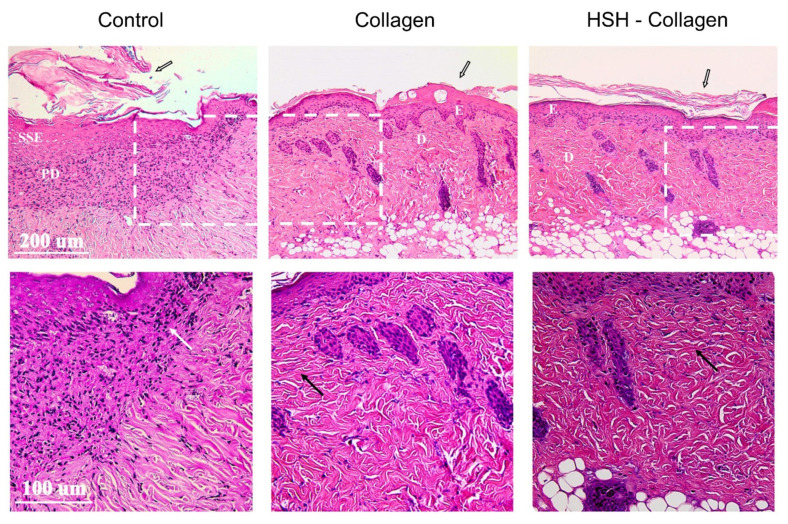
Histological analysis of wound tissues isolated on day 5 post-wounding from groups fed with different diets. Representative photographs of 3 μm sections stained with hematoxylin-eosin. Collagen and HSH-Collagen supplementation showed a more mature dermis compared to the control. E: epidermis, SSE: stratified squamous epithelium, D: dermis, PD: papillary dermis. White arrow shows inflammatory cells within the SSE; black arrows indicate collagenated fibrous tissue; black outlined white arrows illustrate hyperkeratotic epidermal layer. Magnification (×10, scale bar 200 μm) and zoom in panels (×20, scale bar 100 μm).

**Figure 3 marinedrugs-19-00396-f003:**
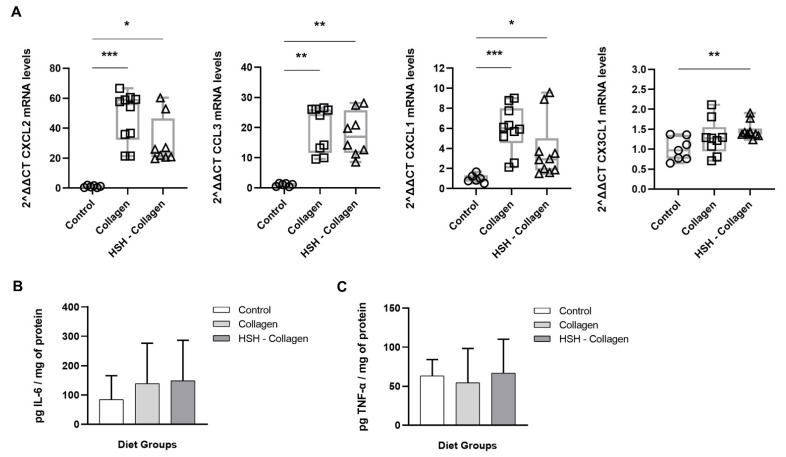
Expression of different chemokines and cytokines at the wound site. (**A**) mRNA expression of CXCL2, CCL3, CXCL1 and CX3CL1 was analyzed by real-time PCR. IL-6 (**B**) and TNF-α (**C**) levels were quantified for each diet group via ELISA on the 5th day post injury. Horizontal bars indicate median values and points indicate individual tissue samples. * *p* < 0.005, ** *p* < 0.01, *** *p* < 0.001.

**Figure 4 marinedrugs-19-00396-f004:**
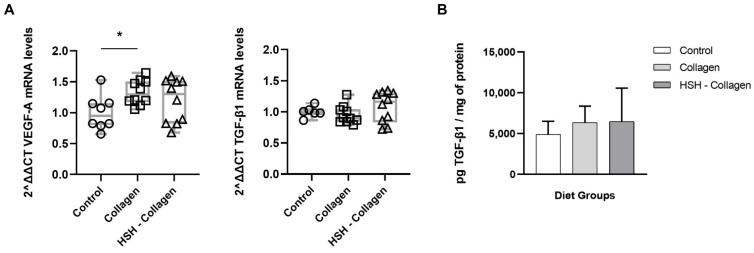
VEGF-A expression was induced by Collagen supplementation. (**A**) Real-time PCR analysis of VEGF-A and TGF-β1 mRNA transcript levels in each diet group. (**B**) Wounds were assessed for the presence of produced TGF-β1 by ELISA. Horizontal bars indicate median values and points indicate individual tissue samples. * *p* < 0.005.

**Figure 5 marinedrugs-19-00396-f005:**
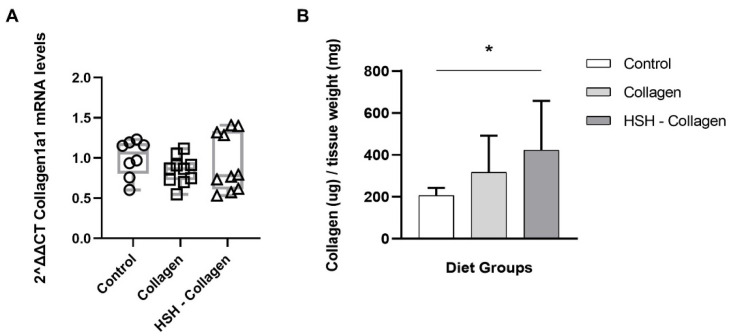
High collagen deposition at the wound site of HSH-Collagen fed mice. (**A**) mRNA expression of Collagen1a1 was analyzed by real-time PCR. (**B**) Quantification of collagen production in wound tissues performed via Sircol™ Soluble Collagen Assay. Horizontal bars indicate median values and points indicate individual tissue samples. * *p* < 0.005.

**Table 1 marinedrugs-19-00396-t001:** Raw material utilized for the production of the fish sidestream-derived protein hydrolysates tested in the cutaneous wound model.

Supplement	Raw Material
HMB	Mackerel Backbones
HMH	Mackerel Heads
HSB	Salmon Backbones
HSH	Salmon Heads
Collagen	Flounder Skin Collagen
HSH-Collagen	50% Salmon Heads + 50% Collagen
Soy (control)	Soy protein

**Table 2 marinedrugs-19-00396-t002:** List of primers used in real-time PCR reactions.

Gene	Sequence
*B-actin*	F: 5′-CATTGCTGACAGGATGCAGAAGG-3′ & R: 5′-TGCTGGAAGGTGGACAGTGAGG-3′
*CXCL1*	F: 5′-CCCAAACCGAAGTCATAGCCA-3′ & R: 5′-CTCCGTTACTTGGGGACACC-3′
*CXCL2*	F: 5′-CGCCCAGACAGAAGTCATAGCCAC-3′ & R: 5′-CGTTGAGGGACAGCAGCCCAG-3′
*CCL3*	F: 5′-GAAGGATACAAGCAGCAGCG-3′ & R: 5′-TTCTCTTAGTCAGGAAAATGACACC-3′
*CX3CL1*	F: 5′-CTACTAGGAGCTGCGACACG-3′ & R: 5′-TGTCGTCTCCAGGACAATGG-3′
*VEGF* *-A*	F: 5′-GTACCTCCACCATGCCAAGT-3′ & R: 5′-ACTCCAGGGCTTCATCGTTA-3′
*TGF-* *β1*	F: 5′-GACACACAGTACAGCAAGGTCC-3′ & R: 5′-CGACCCACGTAGTAGACGATG-3′
*Collagen1a1*	F: 5′-GCTGCACGAGTCACACCG-3′ & R: 5′-GAGGGAACCAGATTGGGGTG-3′

## Data Availability

The data presented in this study are available on request from the corresponding author and can be made publicly available.

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
