# Peer review of "Collagen-Containing Fish Sidestream-Derived Protein Hydrolysates Support Skin Repair via Chemokine Induction"

_marinedrugs, 2021, doi:10.3390/md19070396_

Round 1

Reviewer 1 Report

  1. Section 2.4: The authors described that though HSH-collagen supplementation enhanced chemokine expression, the activity is lesser than Collagen alone. Does this mean that the wound healing activity of HSH-collagen weaker than collagen?
  2. Based on Figure 1B, HMB did not affect on wound size. However, in supplementary figure 1, HMB strongly activate chemokine expression. I think this results account for disaccordance between wound-healing and chemokine. The authors must describe this situation.
  3. Line 191-192: The authors wrote “Overall, a moderate, and not excessive, increase in the expression levels of CC- and CXC-191 chemokine subfamilies may facilitate healing.” I could not find any evidence about moderate modulation of chemokine is good for wound healing.

Author Response

Comment 1: Section 2.4: The authors described that though HSH-collagen supplementation enhanced chemokine expression, the activity is lesser than Collagen alone. Does this mean that the wound healing activity of HSH-collagen weaker than collagen?

Response 1: According to Figures 1B and 2, Collagen and HSH-Collagen supplementation have the same wound healing activity. However, differences in chemokine and VEGF-A expression as well as collagen deposition indicate that these two extracts possibly promote healing through distinct mechanisms. We have specified this issue in the discussion section as follows: “Our findings show that both Collagen and HSH - Collagen supplementation similarly stimulated wound closure, however differences in chemokine and VEGF-A expression as well as collagen deposition indicate that these two extracts possibly promote healing through distinct mechanisms.”.

Comment 2: Based on Figure 1B, HMB did not affect on wound size. However, in supplementary figure 1, HMB strongly activate chemokine expression. I think this results account for discordance between wound-healing and chemokine. The authors must describe this situation.

Response 2: We believe the reviewer meant the supplements HMH or HSB. Both HSB and HMH did not show any effect on wound size, in fact wounds appeared to close slightly slower that the corresponding control. However, wound tissues after supplementation with these two extracts expressed high amounts of chemokines, greater than Collagen and HSH - Collagen that seem to promote wound healing. Therefore, we think that this chemokine overexpression denotes excessive inflammation, which is indicative of a prolonged inflammatory phase and associated with delayed healing. Chemokine expression may facilitate healing, but there are also additional factors that contribute to the skin repair process, promoting wound closure. This is now specified in the discussion section as follows: “An excessive chemokine response during the skin repair process has been linked to overactive inflammation, which could lead to inhibition of neovascularization and subsequent retention in the inflammatory phase, delaying wound closure or creating a hostile environment for the development of a no healing wound”

Comment 3: Line 191-192: The authors wrote “Overall, a moderate, and not excessive, increase in the expression levels of CC- and CXC- chemokine subfamilies may facilitate healing.” I could not find any evidence about moderate modulation of chemokine is good for wound healing.

Response 3: We used the word moderate to illustrate a controlled chemokine response. Too high chemokine expression has been linked to overactive inflammation, which could lead to inhibition of neovascularization and subsequent prolongation of the inflammatory phase, delaying wound closure or creating a hostile environment for the development of a no healing wound. This point is now rephrased in the results section as follows: “On the contrary, diet groups such as HSB and HMH that did not have any positive effect on wound size, in fact these wounds appeared to close slightly slower that the control, showed too high chemokine expression, possibly indicating excessive inflammation at the wound site (Supplementary Figure 1). Overall, a controlled and not excessive increase in the expression levels of CC- and CXC- chemokine subfamilies...” and discussed in the discussion, including related references: “...delaying wound closure or creating a hostile environment for the development of a no healing wound [47–49]”

Reviewer 2 Report

Authors please include a definition of fish sidestreams near the end of the introduction. Consider using wording similar to lines 297-299 in the discussion if this is indeed what fish sidestreams are.

Dose-response of the sidestream materials. The reviewer recognizes why 5% material was added (explained on line 128-130). Was there ever a dose-response considered for the study?

Control used. The soy protein is a plant material; were animal proteins considered as placebo? For example, bovine collagen that was used to generate the standard curve could have been added as the supplement. The study would be much stronger if other sources of collagen had been used in comparison, as well as purchasing, mixing, and administering the materials identified in the supplemental information to see whether this was sufficient to induce the improved wound healing response. 

Literature review. Three other papers specifically used marine fish products to stimulate wound healing; comment on these papers and how they support your results:

  • PMCID: PMC3111176
  • PMCID: PMC4432022
  • PMID: 33061887

Discussion of collagen deposition. A few older studies by Pilcher showed that collagenase expression was required to improve wound healing, presumably by releasing or exposing collagen fragments. This would be useful in your discussion of the mechanism of ingested collagen. Monitoring keratinocyte migration over time in the mouse wounds could be done using an optical coherence tomography or something similar to see whether keratinocytes migrate faster in the treatments.

  • PMCID: PMC2132537
  • PMID: 9710382

Author Response

Comment 1: Authors please include a definition of fish sidestreams near the end of the introduction. Consider using wording similar to lines 297-299 in the discussion if this is indeed what fish sidestreams are.

Response 1: We thank the reviewer for the comment. The definition of fish side streams is included in the revised introduction as follows: A large amount of side streams, around 70–85% of the total weight of catch, is produced by the fish processing industry every year, significantly contributing to environmental pollution [22,24]. Although this raw material is generally considered waste, most of it contains great amount of muscle and connective tissue, high in protein content that may have important health-promoting properties. These proteins can be subjected to enzymatic hydrolysis, which is a mild processing technology that generates protein hydrolysates, a source of bioactive peptides [25–27].

Comment 2: Dose-response of the sidestream materials. The reviewer recognizes why 5% material was added (explained on line 128-130). Was there ever a dose-response considered for the study?

Response 2: We used a single concentration of supplementation since it represents an amount that can be considered a supplement and not regular diet, as noted by the reviewer. Using additional concentrations would require much larger number of experimental animals that would not be easy to justify. Moreover, we did not consider performing a dose response in the study since administration of higher doses would not be considered as supplements and lower doses may not have any action, based on the findings.

Comment 3: Control used. The soy protein is a plant material; were animal proteins considered as placebo? For example, bovine collagen that was used to generate the standard curve could have been added as the supplement. The study would be much stronger if other sources of collagen had been used in comparison, as well as purchasing, mixing, and administering the materials identified in the supplemental information to see whether this was sufficient to induce the improved wound healing response. 

Response 3: This was an exploratory analysis to test if fish sidestream-derived protein hydrolysates had any effect on skin homeostasis, where the aim was to test different sources of fish sidestreams to determine bioactivity. We used soy as a control since soy protein was the one included in the rodent diet the mice were receiving, to exclude the effect of protein in general, regardless its source. Now that we have identified the particular sources of supplements and showed that the collagen-containing supplements stimulate skin repair, in a subsequent study we could further compare with other collagen sources, providing additional value to a potential product.

Comment 4: Literature review. Three other papers specifically used marine fish products to stimulate wound healing; comment on these papers and how they support your results: PMCID: PMC3111176 PMCID: PMC4432022 PMID: 33061887

Discussion of collagen deposition. A few older studies by Pilcher showed that collagenase expression was required to improve wound healing, presumably by releasing or exposing collagen fragments. This would be useful in your discussion of the mechanism of ingested collagen. Monitoring keratinocyte migration over time in the mouse wounds could be done using an optical coherence tomography or something similar to see whether keratinocytes migrate faster in the treatments. PMCID: PMC2132537 PMID: 9710382

Response 4: We thank the reviewer for bringing these papers into our attention. We have now commented on these papers in the discussion as follows: ‘According to earlier studies, type I collagen within the wound triggers the expression of a matrix metalloproteinase, collagenase-1, which in turn facilitate keratinocyte migration during reepithelialization [58,59]. The ingested collagen observed in our study after supplementation with the extracts could promote healing via a similar way of action. In agreement with our results, other studies have shown that oral administration of fish skin-derived extracts improved cutaneous wound healing in rodents by enhancing collagen deposition and angiogenesis. In particular, peptides enzymatically hydrolyzed from the skin of either chum salmon or marine fish have been reported to induce wound healing in rats following cesarean section [60,61]. Zhang et al. also found that oral administration of skin gelatin from chum salmon enhanced wound healing in diabetic rats [62].”

Reviewer 3 Report

The submitted article uses a mouse cutaneous wound model of secondary intention healing to determine the outcome of supplemented diets of hydrolyzed fish byproducts. The author group has a strong publication record utilizing these hydrolyzed proteins for various inflammatory health related outcomes (Colitis, Diabetes, etc). Overall, the results are limited as complete healing is never accessed (7-14 days) and a limited conclusion of increased healing rate for Collagen and HSH-Collagen at day 3 is what is highlighted, though all groups have similar wound size % at day 5. The expression levels from day 5 are interesting, but hard to access what functional outcomes they correlate to since all groups had similar wound size at the time of collection. Conclusions are reasonable, but article would benefit from inclusion of limitations throughout the discussion.     

Points for consideration:

Table 1: Soy should be included in supplement list for control, raw material source is not included in M&M, so unclear

Lines 126-127 – Need to define abbreviations upon first use HSB, HMH, HMB, Collagen, HSH-Collagen (HSH previously defined line 119).

Figure 1 – Wound closure is defined by complete abridgement of the wound edges – where is the measure and quantification of that?

Figure 1C – would be nice to include pictures of all supplement groups for visual comparison.

Figure 2 – Should state clearly from day 5 post-wounding; No (F-label) fibrous connective tissue in histological images (there are arrows)

Figure 3 – What do individual data points represent? Number of mice, number of wounds – it is not clear as M&M section stats n=4-5 mice yet figures have an undisclosed number of individual values – take Cx3CL1 mRNA levels – Control = 7; Collagen = 10; HSH+Collagen = 9. Same issue for Figure 4.

Line 207-8 – VEGF is also important in lymphangiogensis, needs to be included in list of actions. There are 4 mammalian VEGF A-D, with some having opposite down stream effects from others. The measured VEGF in Figure 4A was total (A, B, C & D), VEGF-A, etc? Forward primer looks to be VEGF-A, but not clear on reverse primer. 

Line 239 – “…faster rate of wound closure, especially on day 3” – No, they had faster reduction of wound size% at 3 days – Figure 1B.

Line 331 – What sex were the mice? Age?

Line 339 - Did mice consume supplemented ND at the rate expected 5g/mouse/day? Were body weights monitored? What about water intake as earlier sections reference 'salty taste'? 

Line 340 – Anesthesia dose should be provided for Ketamine/Xylazine

Line 340 – No analgesic information is provided – Did mice receive pain medication for their wounds?

Line 344 – Soy is estrogenic, which is know to reduce inflammation and increase wound healing – ND might have been a better control as it would give more accurate healing profile. Was the soy protein hydrolyzed?

  • Zhou T, Yang Z, Chen Y, Chen Y, Huang Z, You B, Peng Y, Chen J: Estrogen Accelerates Cutaneous Wound Healing by Promoting Proliferation of Epidermal Keratinocytes via Erk/Akt Signaling Pathway. Cell Physiol Biochem 2016;38:959-968. doi: 10.1159/000443048

  • Ashcroft GS, Mills SJ, Lei K, et al. Estrogen modulates cutaneous wound healing by downregulating macrophage migration inhibitory factor. J Clin Invest. 2003;111(9):1309-1318. doi:10.1172/JCI16288

  • Laura Campbell, Elaine Emmerson, Faith Davies, Stephen C. Gilliver, Andre Krust, Pierre Chambon, Gillian S. Ashcroft, Matthew J. Hardman; Estrogen promotes cutaneous wound healing via estrogen receptor β independent of its antiinflammatory activities. J Exp Med 30 August 2010; 207 (9): 1825–1833. doi: https://doi.org/10.1084/jem.20100500

Line 348 - Initial wound was area of a circle with diameter 3mm [A = π (d/2)2] (formula not included), but when wounds became altered shapes, how was the area calculated? Was a software utilized to measure, like ImageJ? 

Line 416 “proteins containing collagen or collagen alone” – Should be ‘containing HSH-collagen or collagen alone’

If day 3 outcomes from HSH-collagen and collagen alone were so profound, would be interesting to look at cytokine/chemokine markers at day 3 rather than day 5. This would also show repeatability of the findings.   

Author Response

Comment: The submitted article uses a mouse cutaneous wound model of secondary intention healing to determine the outcome of supplemented diets of hydrolyzed fish byproducts. The author group has a strong publication record utilizing these hydrolyzed proteins for various inflammatory health related outcomes (Colitis, Diabetes, etc). Overall, the results are limited as complete healing is never accessed (7-14 days) and a limited conclusion of increased healing rate for Collagen and HSH-Collagen at day 3 is what is highlighted, though all groups have similar wound size % at day 5. The expression levels from day 5 are interesting, but hard to access what functional outcomes they correlate to since all groups had similar wound size at the time of collection. Conclusions are reasonable, but article would benefit from inclusion of limitations throughout the discussion.     

Response: We thank the reviewer for the comments. Indeed, the analysis focused on screening different fish sidestream-derived extracts on wound healing and several limitations exist to allow screening of the supplements. According to the suggestion of the reviewer we have included the limitations of the study in the discussion section as follows: “There are several limitations in the study. Analysis of wound closure was performed until day 5, when the wound was not fully healed to allow collection of tissue samples, therefore the timing of complete wound healing was not evaluated. In addition samples were collected to evaluate expression of different factors at the tissue restoration stage and not at the early inflammatory stage of the healing process, restricting the information obtained on the effect of the different supplements. Moreover, the effect on chemokine expression was analyzed at the mRNA level, not allowing identification of a potential effect at the post-transcriptional level. Nevertheless, our findings clearly support the effect of collagen containing fish sidestream-derived supplements on skin homeostasis.”

Points for consideration:

Comment 1: Table 1: Soy should be included in supplement list for control, raw material source is not included in M&M, so unclear

Response 1: We thank the reviewer for bringing this to our attention. We have now included soy in Table 1 and we added related information in the Materials and Methods section as follows: “A concentration of 5% w/w in normal chow diet (4RF21, Mucedola, Settimo Milanese, MI, Italy) was used for each extract. Since these extracts contain high amount of protein and the fact that 84% of the total protein in normal diet is soy, we supplemented normal diet with 5% w/w of 100% pure soy protein and utilized it as the control diet group.”

Comment 2: Lines 126-127 – Need to define abbreviations upon first use HSB, HMH, HMB, Collagen, HSH-Collagen (HSH previously defined line 119).

Response 2: We have now defined the abbreviation upon first use of each extract.

Comment 3: Figure 1 – Wound closure is defined by complete abridgement of the wound edges – where is the measure and quantification of that?

Response 3: Since the induced wound had circle-like shape, abridgement occurred circumferential but closure was not uniform. Therefore, the quantification of wound edge abridgement is most reliably reflected from the area of the wound. At day 5 the wound size was reduced to around 20-30% of the original area, and therefore the experiment was terminated to allow acquiring wound tissue. This is now discussed as a limitation in the discussion section: “There are several limitations in the study. Analysis of wound closure was performed until day 5, when the wound was not fully healed to allow collection of tissue samples, therefore the timing of complete wound healing was not evaluated. In addition samples were collected to evaluate expression of different factors at the tissue restoration stage and not at the early inflammatory stage of the healing process, restricting the information obtained on the effect of the different supplements. Nevertheless, our findings clearly support the effect of collagen containing fish sidestream-derived supplements on skin homeostasis.”

Comment 4: Figure 1C – would be nice to include pictures of all supplement groups for visual comparison.

Response 4: We thank the reviewer for the suggestion. We have now included representative pictures of all supplement groups in Figure 1C.

Comment 5: Figure 2 – Should state clearly from day 5 post-wounding; No (F-label) fibrous connective tissue in histological images (there are arrows).

Response 5: We have now included the information on the day, as suggested. Since fibrous tissue is part of dermis, we believe that the arrows clearly indicate collagenated fibrous tissue, therefore the F-label has been deleted.

Comment 6: Figure 3 – What do individual data points represent? Number of mice, number of wounds – it is not clear as M&M section stats n=4-5 mice yet figures have an undisclosed number of individual values – take Cx3CL1 mRNA levels – Control = 7; Collagen = 10; HSH+Collagen = 9. Same issue for Figure 4.

Response 6: We apologize for the confusing information. We have now clearly indicated that the individual data points represent number of wound tissues analyzed from 4-5 mice/group. Points are missing in some samples due to RNA purification problems. The information is now included in the figure legends.

Comment 7: Line 207-8 – VEGF is also important in lymphangiogensis, needs to be included in list of actions. There are 4 mammalian VEGF A-D, with some having opposite down stream effects from others. The measured VEGF in Figure 4A was total (A, B, C & D), VEGF-A, etc? Forward primer looks to be VEGF-A, but not clear on reverse primer. 

Response 7: Indeed, we used Primer-BLAST to design this set of primers for VEGF-A, which is now indicated. We also commented on its action in lymphangiogenesis in the discussion section as follows: Collagen supplementation may promote healing via VEGF-A expression, which is known to enhance the formation of lymphatic capillaries in cutaneous wound repair [50].

Comment 8: Line 239 – “…faster rate of wound closure, especially on day 3” – No, they had faster reduction of wound size% at 3 days – Figure 1B.

Response 8: We thank the reviewer for the comment. this is now corrected to: “.. reduction of wound size on the third day post injury”

Comment 9: Line 331 – What sex were the mice? Age?

Response 9: We used male mice, 6-8 weeks of age. The information is now included in the Materials and Methods section.

Comment 10: Line 339 - Did mice consume supplemented ND at the rate expected 5g/mouse/day? Were body weights monitored? What about water intake as earlier sections reference 'salty taste'? 

Response 10: Since these extracts contain high amount of protein and the fact that 84% of the total protein in normal chow diet is soy, we supplemented the control group with 5% w/w of pure soy protein and utilized it as the control diet group. In addition, we have previously showed that mice fed with the mackerel heads-based hydrolysate (HMH) consumed more water compared to the other diet groups. Yet no difference was observed on daily food consumption. This information is in the following publication: “Daskalaki, M.G.; Axarlis, K.; Aspevik, T.; Orfanakis, M.; Kolliniati, O.; Lapi, I.; Tzardi, M.; Dermitzaki, E.; Venihaki, M.; Kousoulaki, K.; et al. Fish Sidestream-Derived Protein Hydrolysates Suppress DSS-Induced Colitis by Modulating Intestinal Inflammation in Mice. Mar. Drugs 2021, 1–16.” We have now included this information and the related citation in the results section as follows: “This was also evident by monitoring the daily water consumption as we have previously shown that mice fed with the hydrolysate based on mackerel heads, HMH, consumed more water compared to the other diet groups; yet no difference was observed on daily food consumption [29].”

Comment 11: Line 340 – Anesthesia dose should be provided for Ketamine/Xylazine

Response 11: We used a mixture of 120 mg/kg of body weight ketamine, 8 mg/kg xylazine and 1 ug/Kg fentanyl. Mice were kept sedated for about 30 minutes. We included this information in the Materials and Methods section

Comment 12: Line 340 – No analgesic information is provided – Did mice receive pain medication for their wounds?

Response 12: We apologize for omitting this information. Indeed, analgesia was used by administering 1 ug/kg of body weight fentanyl. This information is now included in the Materials and Methods section.

Comment 13: Line 344 – Soy is estrogenic, which is know to reduce inflammation and increase wound healing – ND might have been a better control as it would give more accurate healing profile. Was the soy protein hydrolyzed?

  • Zhou T, Yang Z, Chen Y, Chen Y, Huang Z, You B, Peng Y, Chen J: Estrogen Accelerates Cutaneous Wound Healing by Promoting Proliferation of Epidermal Keratinocytes via Erk/Akt Signaling Pathway. Cell Physiol Biochem 2016;38:959-968. doi: 10.1159/000443048
  • Ashcroft GS, Mills SJ, Lei K, et al. Estrogen modulates cutaneous wound healing by downregulating macrophage migration inhibitory factor. J Clin Invest. 2003;111(9):1309-1318. doi:10.1172/JCI16288
  • Laura Campbell, Elaine Emmerson, Faith Davies, Stephen C. Gilliver, Andre Krust, Pierre Chambon, Gillian S. Ashcroft, Matthew J. Hardman; Estrogen promotes cutaneous wound healing via estrogen receptor β independent of its antiinflammatory activities. J Exp Med30 August 2010; 207 (9): 1825–1833. doi: https://doi.org/10.1084/jem.20100500

Response 13: For this study we aimed to test whether diet supplementation with fish-derived protein supplements affected wound healing process. The control group included soy protein since the major protein source in rodent diet is soy (Cat. No:4RF21, Mucedola, Settimo Milanese, MI, Italy, which contains 18.2% protein of which 84% is soy protein). We, thus, believe that a control without a protein supplement will not be the appropriate control because one could argue that any effect could be due to the increased protein content of the supplement and not specific to the particular supplement. The soy protein was not hydrolyzed. Soy protein indeed has estrogenic properties but an addition of 5% in the total feed that already contains soy protein may have an effect, which could potentially mask the positive effect of our supplements. We have now included this information in the discussion section as follows: “Control mice were fed with normal diet plus 5% w/w soy protein given that the fish sidestream-derived extracts contain high amount of protein and the fact that it soy represents 84% of total protein in normal chow diet. However, soy is estrogenic and some studies have reported that estrogen reduces inflammation and affects wound healing, which could potentially mask the positive effects of the fish sidestream-derived supplements [34–36].”

Comment 14: Line 348 - Initial wound was area of a circle with diameter 3mm [A = π (d/2)2] (formula not included), but when wounds became altered shapes, how was the area calculated? Was a software utilized to measure, like ImageJ? 

Comment 14: Indeed, we used the ImageJ software for calculating the wound area. This information is now included in the Materials and Methods section as follows: “Changes in wound size were calculated using ImageJ software and expressed as the percentage of the initial wound area.”

Comment 15: Line 416 “proteins containing collagen or collagen alone” – Should be ‘containing HSH-collagen or collagen alone’

Comment 15: We have now corrected the statement accordingly.

Comment 16: If day 3 outcomes from HSH-collagen and collagen alone were so profound, would be interesting to look at cytokine/chemokine markers at day 3 rather than day 5. This would also show repeatability of the findings.

Comment 16: We agree with the reviewer that such an experiment would provide more in depth information on the different stages of the wound healing process. Nevertheless, in this study we wanted first to test a series of protein hydrolysates from fish sidestreams and whether they have an effect on wound healing. The study focused on screening different supplements rather than analyzing in depth the effect of the mechanism, which could be the focus of a future study on the most effective fish-derived supplement.

Reviewer 4 Report

In this paper the Authors investigate and analyse d the expression and production of factors 28

that participate in different stages of the skin repair process, testing fish side- stream-derived collagen alone (Collagen), or in combination with a protein hydrolysate derived from salmon heads (HSH), in a model of cutaneous wound healing.

They conclude that use of collagen-containing dietary supplements of marine origin may promote skin repair.

This work is interesting although it is not yet acceptable in this form for a possible publication. The authors have to answer to some critic question about the contents of the paper.

Belove the comments to authors have been reported

Introduction section

-The introduction is lengthy and sometimes redundant on the concept of malnutrition. I suggest streamlining it by better highlighting the purpose of this study.

Results section

Selection of fish sidestream-derived protein hydrolysates:

-Lines 101-107 (including ref 26-28): I think this whole part in the paragraph fits better with the results paragraph of the discussion. The authors in the paragraph can offer only the experimental data obtained from this type of research. The same goes for the following sentences. If they are not results obtained from analyzes conducted by the authors, it makes no sense to report them as results.

Accelerated wound healing is observed in Collagen and HSH - Collagen fed mice.

Th authors declare that mice receiving as supplement Collagen or HSH - Collagen was significantly reduced in the third post-wounding day compared to the corresponding control group. However, this conclusion is not reliable because it is not adequately supported by the statistical analysis, as the authors do not specify on how many tests they performed the statistical analysis. This evident difference in the restorative capacity that the authors claim to observe does not show honestly from the images

Collagen production is increased in HSH - Collagen supplemented group.

In Figure 5 (Panel B) although the authors declare that collagen deposition at the wound bed was enhanced in both groups there does not appear to be a statistically significant difference at all.

Finally, in general terms, the determination of the cytokines carried out exclusively with RT-PCR is not sufficient to determine if there is a real difference in the expression of the relative proteins in the different conditions tested. I suggest to the authors to verify this aspect, as it is not certain that there is at the post-transcriptional level the same situation observed in the regulation of the mRNA expression of chemokines.

Discussion section

Before to conclude that results obtained in this work propose that the extracts, Collagen and HSH - Collagen, have a positive role in the reparative and regenerative process of wound healing when used as dietary supplement, the authors have to carefully carry out an accurate statistical analysis and highlight this real reparative capacity which is not evident at all eloquently in the images that have been presented in this work. Discussion becomes the discussion becomes purely speculative at this point

Materials and Methods section

Preparation of fish side stream-derived nutritional supplements:

- the authors in addition to indicating the reference, the authors must include a brief description of the experimental procedures used.

Wounding

-Indicate the number of animals employed for the experiments.

Tissue processing and histological analysis

-Line 365: Specify the percentage of the fixative used

ELISA

-Please report the measure units used for the cytokine quantification.

Author Response

Comment: In this paper the Authors investigate and analyze d the expression and production of factors 28

that participate in different stages of the skin repair process, testing fish side- stream-derived collagen alone (Collagen), or in combination with a protein hydrolysate derived from salmon heads (HSH), in a model of cutaneous wound healing. They conclude that use of collagen-containing dietary supplements of marine origin may promote skin repair. This work is interesting although it is not yet acceptable in this form for a possible publication. The authors have to answer to some critic question about the contents of the paper.

Response 1: We thank the reviewer for the positive view of our work.

Below the comments to authors have been reported

Introduction section

Comment 1: -The introduction is lengthy and sometimes redundant on the concept of malnutrition. I suggest streamlining it by better highlighting the purpose of this study.

Response 1: We have now shortened the introduction accordingly, trying to better highlight the purpose of our study.

Results section

Selection of fish sidestream-derived protein hydrolysates:

Comment 2: -Lines 101-107 (including ref 26-28): I think this whole part in the paragraph fits better with the results paragraph of the discussion. The authors in the paragraph can offer only the experimental data obtained from this type of research. The same goes for the following sentences. If they are not results obtained from analyzes conducted by the authors, it makes no sense to report them as results.

Response 2: We thank the reviewer for the suggestion. We have now moved this paragraph to the introduction section. In section 2.1, we introduce the extracts used in our study and provide information regarding the chemical and nutritional analysis performed (supplementary tables), highlighting the extracts that contain high amount of nutrients which are important for wound healing. The title of section 2.1 has been changed to “Chemical and nutritional analysis of fish sidestream-derived protein hydrolysates”.

Accelerated wound healing is observed in Collagen and HSH - Collagen fed mice.

Comment 3: The authors declare that mice receiving as supplement Collagen or HSH - Collagen was significantly reduced in the third post-wounding day compared to the corresponding control group. However, this conclusion is not reliable because it is not adequately supported by the statistical analysis, as the authors do not specify on how many tests they performed the statistical analysis. This evident difference in the restorative capacity that the authors claim to observe does not show honestly from the images.

Response 3: In response to the comment of Reviewer 3 and this reviewer we have now included photographs from all supplemented groups and used different representative pictures for groups control and HSH – Collagen to better emphasize the difference observed on day 3 according to Figure 1B. To analyze the time-course curve, we performed two-way ANOVA followed by Sidak’s post hoc test. We also employed Bonferroni and Holm-Sidak post hoc tests and all tests showed the same results. This information is now included in the figure legend and the Materials and Methods section as follows: “two-way ANOVA followed by Sidak’s post hoc test was employed to analyze the time-course curve for the wound healing process. Results were confirmed using Bonferroni and Holm-Sidak post hoc test.”

Collagen production is increased in HSH - Collagen supplemented group.

Comment 4: In Figure 5 (Panel B) although the authors declare that collagen deposition at the wound bed was enhanced in both groups there does not appear to be a statistically significant difference at all.

Response 4: We apologize for the error, which is now corrected mentioning that collagen deposition was increased in the HSH-collagen group.

Comment 5: Finally, in general terms, the determination of the cytokines carried out exclusively with RT-PCR is not sufficient to determine if there is a real difference in the expression of the relative proteins in the different conditions tested. I suggest to the authors to verify this aspect, as it is not certain that there is at the post-transcriptional level the same situation observed in the regulation of the mRNA expression of chemokines.

Response 5: The majority of the published papers that investigate wound healing in different settings, predominantly study chemokine expression at the mRNA level. As we wanted to screen different chemokines, we focused on the mRNA expression levels to explore the first signals that activate the cellular response in the wounded tissue. Chemokines are primarily regulated at the transcriptional level, even though post-transcriptional regulation of some chemokines also occurs at the level of mRNA stability, translation, and release. We have now included this information in the discussion section where we highlight the limitations of our study as follows: “Moreover, the effect on chemokine expression was analyzed at the mRNA level, not allowing identification of a potential effect at the post-transcriptional level. Nevertheless, our findings clearly support the effect of collagen containing fish sidestream-derived supplements on skin homeostasis.”

Discussion section

Comment 6: Before to conclude that results obtained in this work propose that the extracts, Collagen and HSH - Collagen, have a positive role in the reparative and regenerative process of wound healing when used as dietary supplement, the authors have to carefully carry out an accurate statistical analysis and highlight this real reparative capacity which is not evident at all eloquently in the images that have been presented in this work. Discussion becomes the discussion becomes purely speculative at this point.

Response 6: According to comment 3 of the reviewer, we employed different post hoc tests and they showed that there is a statistically significant difference between the control and Collagen or HSH - Collagen on day 3. We also added representative photographs of all supplements and utilized different representative photos for control and HSH - Collagen in Figure 1C to better highlight the difference in wound size reduction between the diet groups to support our conclusions.

Materials and Methods section

Comment 7: Preparation of fish side stream-derived nutritional supplements:

- the authors in addition to indicating the reference, the authors must include a brief description of the experimental procedures used.

Response 7: We thank the reviewer for the suggestion. We have now added a brief description of the procedure involved in preparation of the supplements as follows: “Briefly, raw materials were milled and combined with tap water. For the enzymatic hydrolysis, mixtures were stirred vigorously before adding the enzyme (10 U/g protein). After one-hour-hydrolysis at 55 °C, the temperature was raised at 90 °C to terminate the enzyme activity. The temperature was then lowered to 60 °C and the hydrolysate was placed into a 3-phase separating centrifuge. Enzyme inactivation at 90 °C was followed and the liquid phase was separated into the oil and aqueous phase. The aqueous phase was further filtered and concentrated on a 4-stage falling film evaporator before spray-drying. The resulted purified hydrolysates were hygroscopic, white to off-white, fluffy powders.”

Wounding

Comment 8: -Indicate the number of animals employed for the experiments.

Response 8: We have used 4-5 mice per group. The information is now included in the Materials and Methods section.

Tissue processing and histological analysis

Comment 9: -Line 365: Specify the percentage of the fixative used

Response 9: Tissues were fixed in 10% formalin. This is now indicated in the Materials and Methods section.

ELISA

Comment 10: -Please report the measure units used for the cytokine quantification.

Comment 10: The measurement units used for quantification were in picogram of each cytokine (TGF-β1/IL-6/TNF-α) normalized per milligram of tissue protein for each sample. This information is now included in the Materials and Methods section.

Round 2

Reviewer 3 Report

All comments from previous round of review have been appropriately addressed and included in the current version of the manuscript. 

Reviewer 4 Report

The authors have modified the manuscript in accordance with the suggestions of the reviewers, significantly improving its overall quality. no other comments to add.